# Foundation Models as Class-Incremental Learners for Dermatological Image Classification

Mohamed Elkhayat[1*], Mohamed Mahmoud[1*],
Jamil Fayyad[2†], and Nourhan Bayasi[3†]

[1] Cairo University, Giza, Egypt
[2] University of Victoria, Victoria, BC, Canada
[3] University of British Columbia, Vancouver, BC, Canada
{Mohammed.Khayyat02, muhammad.mahmoud01}@eng-st.cu.edu.eg

**Abstract.** Class-Incremental Learning (CIL) aims to learn new classes over time without forgetting previously acquired knowledge. The emergence of foundation models (FM) pretrained on large datasets presents new opportunities for CIL by offering rich, transferable representations. However, their potential for enabling incremental learning in dermatology remains largely unexplored. In this paper, we systematically evaluate frozen FMs pretrained on large-scale skin lesion datasets for CIL in dermatological disease classification. We propose a simple yet effective approach where the backbone remains frozen, and a lightweight MLP is trained incrementally for each task. This setup achieves state-of-the-art performance without forgetting, outperforming regularization, replay, and architecture-based methods. To further explore the capabilities of frozen FMs, we examine zero-training scenarios using nearest-mean classifiers with prototypes derived from their embeddings. Through extensive ablation studies, we demonstrate that this prototype-based variant can also achieve competitive results. Our findings highlight the strength of frozen FMs for continual learning in dermatology and support their broader adoption in real-world medical applications. Our code and datasets are available here.

**Keywords:** Class-Incremental Learning · Continual Learning · Foundation Models · Dermatological Image Classification · Dermatology

## 1 Introduction

Real-world clinical applications rarely offer the luxury of independent and identically distributed (i.i.d.) data [15]. In dermatology, new disease classes or imaging variations may appear gradually as data is collected over time from different sources. Conventional models trained in a static setting often fail under these changing conditions [16], showing a sharp drop in performance on previously learned tasks when updated with new data; a problem known as *catastrophic*

---

[*] Equal contribution.
[†] Equal supervision.

*forgetting* [26,4]. Continual learning (CL) aims to address this challenge by allowing models to learn new information while preserving past knowledge. Several setups within CL include Class-Incremental Learning (CIL) is one of the most challenging. In CIL, new classes are introduced over time, and the model must learn them without access to data from earlier tasks, making this a relevant setting in clinical workflows where storing or replaying patient data is often restricted due to privacy and ethical concerns.

To avoid forgetting without storing or replaying old patient data [11], researchers have explored regularization- and architecture-based strategies. Regularization methods penalize changes to important parameters [32], while architecture-based approaches expand the model or allocate task-specific components [36,5,10]. While effective in controlled settings, these methods face key limitations in clinical practice: regularization requires reliable importance estimates, which are often difficult to obtain in data-scarce environments, and architecture-based techniques introduce complexity and memory overhead. In contrast, foundation models (FM) trained on large-scale datasets have reshaped the landscape of CIL, offering robust, transferable features that generalize well with minimal fine-tuning [12]. Recent work in natural image domains shows that simply leveraging frozen FMs can significantly boost performance and reduce forgetting [19,20].

Motivated by these findings, we turn to dermatology and ask: Can frozen FMs pretrained on large-scale dermatological data support CIL for skin lesion classification, or are specialized CL methods still necessary? To answer this, we present the first comprehensive evaluation of frozen dermatology FMs for continual skin lesion classification. Our setup is deliberately simple: the backbone remains frozen, and a lightweight MLP classifier is incrementally trained for each task. Surprisingly, this approach outperforms prior CIL methods, including regularization, replay, and architectural techniques, without requiring any fine-tuning. We also explore zero-training setups using prototype-based classifiers derived from FM embeddings, and through extensive ablation studies, demonstrate that variations of this method can significantly outperform existing approaches. Our results suggest that future dermatology-based CL research should start with FMs, rather than designing methods from scratch.

## 2   Related Work

**Class-Incremental Learning for Medical Imaging.**  CIL has recently received growing attention in medical imaging, driven by the need for models that can learn new disease categories without forgetting prior knowledge, all while preserving patient privacy. This has spurred data-free methods that synthesize prior class representations instead of storing raw images. For example, Ayromlou et al. [1] use gradient inversion and novel loss functions to preserve class discriminability. Bayasi et al. [7,6,9] introduce a pruning-based approach that builds independent subnetworks to eliminate forgetting and support fair and generalizable CL. Others rely on regularization: Chee et al. [13] expand network

capacity while retaining prior knowledge, and Chen et al. [14] use contrastive learning and distillation for class- and domain-incremental segmentation.

**Foundation Models in Continual Learning.** Traditional `CL` methods often train feature extractors from scratch, making them vulnerable to catastrophic forgetting. Recent work has shown that leveraging frozen `FMs` can improve both stability and efficiency. In vision, methods like DualPrompt [30] and L2P [31] use prompt tuning on frozen backbones, while Janson et al. [19] showed that simple classifiers on frozen features can rival or outperform complex methods. In medical imaging, Yang et al. [34] used fixed encoders with Gaussian mixtures, Zhang et al. [35] introduced adapter modules, and Bayasi et al. [8] leveraged frozen model ensembles. Yet, the role of `FMs` in continual dermatology classification remains unexplored, leaving an important gap in the field.

## 3  Methodology

### 3.1  Problem Setup

Let $\mathcal{D} = \{(\mathbf{x}_i, y_i)\}_{i=1}^{N}$ denote a dataset of skin lesion images, where $\mathbf{x}_i \in \mathbb{R}^{H \times W \times C}$ is an input image and $y_i \in \{1, 2, \ldots, C\}$ is its corresponding class label. In the class-incremental learning (`CIL`) setup, the complete set of classes $\mathcal{C} = \{1, 2, ..., C\}$ is partitioned into $T$ disjoint subsets, $\mathcal{C}_1, \mathcal{C}_2, ..., \mathcal{C}_T$, such that new classes are introduced sequentially over $T$ tasks. At each time step $t \in \{1, \ldots, T\}$, the model receives access only to a task-specific dataset $\mathcal{D}_t = \{(\mathbf{x}_i, y_i) \mid y_i \in \mathcal{C}_t\}$. No access is granted to prior task data $\mathcal{D}_{<t}$, and storage of past examples is not allowed. The model must update its classification capabilities to accommodate new classes in $\mathcal{C}_t$ while preserving performance on all previously learned classes $\mathcal{C}_{<t} = \bigcup_{j=1}^{t-1} \mathcal{C}_j$.

Let $\mathcal{F}_\theta$ be a dermatology `FM` with frozen parameters $\theta$, pretrained on a large-scale skin lesion images. The parameters $\theta$ remain fixed and are never updated during the continual learning (`CL`) process. For an input image $\mathbf{x}$, the model produces a feature embedding:

$$\mathbf{z} = \mathcal{F}_\theta(\mathbf{x}) \in \mathbb{R}^d \quad .$$

Our goal is to evaluate two `CL` baselines built on top of these frozen embeddings. The first one uses an MLP-based classifier, where a lightweight multi-layer perceptron is incrementally trained on top of the frozen embeddings for each new task. The second one adopts a prototype-based nearest-mean classifier (NMC), which requires no training. Instead, it computes a mean feature vector (prototype) for each class using the frozen features of the labeled training samples.

### 3.2  Baseline 1: MLP-Based Class-Incremental Learning

In this baseline, we keep the `FM` frozen and train a lightweight MLP classifier incrementally across tasks.

**3.2.1. Training Phase.** At each task $t$, a new MLP head $h_t : \mathbb{R}^d \rightarrow \mathbb{R}^{|\mathcal{C}_t|}$ is trained on the frozen embeddings from $\mathcal{D}_t$. The MLP has two hidden layers with ReLU activation and a softmax output:

$$h_t(\mathbf{z}) = \text{Softmax}\left(W_3 \cdot \text{ReLU}\left(W_2 \cdot \text{ReLU}(W_1\mathbf{z} + \mathbf{b}_1) + \mathbf{b}_2\right) + \mathbf{b}_3\right) \quad .$$

To support all seen classes, we concatenate the outputs of all MLPs learned up to task $t$: $h(\mathbf{z}) = \text{Concat}(h_1(\mathbf{z}), \ldots, h_t(\mathbf{z}))$.

**3.2.2. Inference Phase.** At test time, input image $\mathbf{x}$ is passed through the frozen encoder and all MLP heads. The final prediction is made by taking the class with the highest probability across all tasks: $\hat{y} = \arg\max_{c \in \mathcal{C}_{\leq t}} h(\mathcal{F}_\theta(\mathbf{x}))_c$.

### 3.3    Baseline 2: Prototype-Based Nearest Mean Classifier (NMC)

This baseline avoids training by using class-wise mean embeddings (prototypes) computed from frozen features.

**3.3.1. Training Phase.** For each class $c \in \mathcal{C}_t$, we compute a class prototype $\mu_c$ by averaging the frozen embeddings of all class-wise training samples in $\mathcal{D}_t$:

$$\mu_c = \frac{1}{|\mathcal{D}_c|} \sum_{(\mathbf{x}_i, y_i) \in \mathcal{D}_t, y_i = c} \mathcal{F}_\theta(\mathbf{x}_i) \quad .$$

These prototypes are stored in a memory bank: $\mathcal{M}_t = \{\mu_c \mid c \in \mathcal{C}_t\}$.

**3.3.2. Inference Phase.** Given a test image $\mathbf{x}$, we extract its embedding $\mathbf{z} = \mathcal{F}_\theta(\mathbf{x})$, then classify it by assigning the label of the nearest prototype across all seen classes: $\hat{y} = \arg\min_{c \in \mathcal{C}_{\leq t}} \|\mathbf{z} - \mu_c\|_2$.

## 4    Experiments and Results

We evaluate our two FM-based CL baselines on the task of skin lesion classification under the CIL setting, where new sets of classes are introduced sequentially without access to previous data. Details are given next.

### 4.1    Experimental Setup

**Datasets.** Our experiments are conducted on three publicly available dermatology datasets: HAM10000 (HAM) [28], Dermofit (DMF) [2], and Derm7pt (D7P) [21], comprising 10,015 1,211, and 963 dermoscopic images, respectively. These datasets were collected from diverse clinical sources and span a subset of seven skin lesion classes. To simulate a CIL scenario, each dataset is partitioned into $T$ tasks with mutually exclusive class labels. We adopt the dataset splits and experimental protocol from [8] to ensure fair and consistent comparison.

**Implementation Details.** We evaluate our baselines using two dermatology-based FMs: the Google Derm model [18], a publicly released FM trained on over 400 skin conditions, and PanDerm [33], a large-scale FM pre-trained on millions

of clinical and dermoscopic dermatology images. Both models are used as frozen feature extractors throughout the continual learning process, with no fine-tuning. The MLP-based classifier is trained using the Adam optimizer (learning rate 0.001, batch size 200) with cross-entropy loss. Training runs for up to 200 epochs per task, with early stopping based on validation accuracy to mitigate overfitting.

**Reference Methods and Competitors.** We compare our baselines with three standard reference methods: SINGLE, which trains separate models for different tasks and deploys a specific model for each task during inference; JOINT, which aggregates the data from all tasks as a consolidated dataset to jointly train a single model (aka. multitask learning); and SeqFT, which fine-tunes a single model on the current task, without any countermeasure to forgetting. We compare against several CL competitors, including two regularization-based methods: EWC [22] and LwF [23]; two generative-based method: DGM [25] and BIR [29]; two replay-based method: iCaRL [27] and RM [3] and a frozen pretrained model-based method: Continual-Zoo [8].

**Evaluation Metrics.** We report the balanced accuracy (**BAAC**), which accounts for class imbalance by averaging the recall across all classes, ensuring that each class contributes equally to the final score. Also, we report the forgetting measure (**F**), which quantifies how much the model forgets previously learned tasks: $\mathbf{F} = \frac{1}{T-1} \sum_{i=1}^{T-1} \max_{k \in \{1, ..., T-1\}} a_{k,i} - a_{T,i}$ , where $a_{k,i}$ is the accuracy on task $i$ after training on task $k$, and $a_{T,i}$ is the final accuracy on task $i$ after training on all $T$ tasks. A higher value of **F** indicates more forgetting.

## 4.2   Results and Analysis

**Main Results.** Table 1 summarizes the performance of our two FM-based baselines across three skin lesion benchmarks. Our MLP-based models (Google Derm and PanDerm) consistently achieve state-of-the-art balanced accuracy (BAAC) while exhibiting zero forgetting ($\mathbf{F} = 0$), outperforming all existing CL methods including regularization, replay, and architecture-based approaches. On the HAM dataset, PanDerm with MLP achieves a BAAC of 92.25%, surpassing even the upper-bound SINGLE model (88.35%) and strongly outperforming replay-based methods like RM. Similar trends are observed on DMF, where PanDerm with MLP reaches 93.11%, exceeding the best non-foundation continual learning method, Continual-Zoo, by over 20 percentage points. On the D7P dataset, PanDerm again leads with a BAAC of 77.80%, outperforming all methods.

Interestingly, our NMC-based FM baselines, particularly with Google Derm, achieve comparable, and sometimes superior, results relative to other competing techniques. For example, NMC with Google Derm on D7P surpasses all CL methods and even JOINT. However, their performance lag behind their MLP counterparts, reflecting their inability to adapt to complex or overlapping class boundaries typical of medical imaging and skin lesion data. By contrast, MLPs can learn more flexible decision boundaries in the embedding space, better leveraging the rich features of the frozen FM. These results suggest that the choice of classifier plays an important role in realizing the full potential of frozen founda-

**Table 1.** Performance evaluation of our FM-based baselines and existing methods on three skin lesion classification benchmarks in the CIL setting. Numbers in parentheses next to replay- or generative-based methods indicate the number of stored or generated samples per old class, respectively. Green and blue cells denote the best and second-best results, respectively.

| Method | HAM | | DMF | | D7P | |
|---|---|---|---|---|---|---|
| | **BAAC** (↑) | **F** (↓) | **BAAC** (↑) | **F** (↓) | **BAAC** (↑) | **F** (↓) |
| Reference Methods | | | | | | |
| SINGLE | 88.35 | - | 85.01 | - | 73.74 | - |
| JOINT | 82.13 | - | 80.66 | - | 68.32 | - |
| SeqFT | 51.54 | 50.76 | 37.21 | 55.25 | 36.51 | 52.18 |
| Competing Methods | | | | | | |
| EWC | 59.84 | 32.29 | 50.86 | 43.72 | 42.73 | 38.51 |
| LWF | 61.22 | 33.70 | 49.87 | 40.69 | 40.67 | 35.66 |
| DGM | 75.97 | 19.27 | 64.99 | 25.47 | 61.24 | 22.38 |
| BIR | 74.39 | 17.85 | 61.47 | 19.18 | 62.90 | 19.65 |
| iCaRL (50) | 70.80 | 18.44 | 64.32 | 20.17 | 60.84 | 24.78 |
| iCaRL (100) | 73.27 | 14.97 | 68.49 | 18.27 | 63.72 | 19.28 |
| RM (50) | 73.61 | 16.83 | 63.73 | 16.73 | 63.05 | 22.57 |
| RM (100) | 76.32 | 15.92 | 70.14 | 15.22 | 65.87 | 20.17 |
| Continual-Zoo | 78.15 | 11.09 | 72.51 | 14.21 | 68.04 | 17.58 |
| Ours (Baseline 1: FM with MLP) | | | | | | |
| Google Derm | 89.26 | 0 | 91.35 | 0 | 74.59 | 0 |
| PanDerm | **92.25** | **0** | **93.11** | **0** | **77.80** | **0** |
| Ours (Baseline 2: FM with NMC) | | | | | | |
| Google Derm | 64.75 | 0 | 67.56 | 0 | 68.74 | 0 |
| PanDerm | 57.95 | 0 | 49.27 | 0 | 44.51 | 0 |

tion features in the CIL setting. Motivated by this, we explore enhancements for NMC-based models in the subsequent ablation studies.

**Ablation Studies.** We conduct ablation studies to understand design choices in our approach: (1) exploring variants of the NMC classifier, and (2) evaluating the impact of replacing dermatology-specific FMs with general-purpose alternatives.

**1. NMC Classifier Variants.** Table 2 reports the performance of several variants of the base NMC evaluated on HAM, DMF and D7P benchmarks. We begin with a straightforward yet effective enhancement: applying $\ell_2$ normalization to embeddings prior to centroid computation. This standardization consistently improves accuracy by better aligning the embedding space for distance-based decisions. For example, on DMF with Google Derm, accuracy increases from 67.56% to 69.46%. Next, we explore projection-based variants that transform embeddings before classification. Random projection [24] into a higher-dimensional Euclidean space yields limited gains; however, when combined with normalization, modest improvements are observed; for instance, PanDerm accuracy on DMF increases from 49.57% to 54.38%. The most substantial improvements arise from our learnable hyperbolic projection [17], which maps embeddings onto a hyperbolic manifold whose parameters are optimized during training. This projection explicitly captures hierarchical and relational structures among classes, adapting the embedding geometry to improve clustering and distance-based decision boundaries. The impact is significant: on HAM, accuracy rises from 64.75% to 81.41% with the Google Derm model and from 57.95% to 80.24% with PanDerm. Further, combining the hyperbolic projection with normalization boosts

**Table 2.** Performance evaluation (balanced accuracy %) of different variations of the NMC classifier across three skin lesion classification benchmarks. Green cells denote the best results.

| Method | HAM | | DMF | | D7P | |
|---|---|---|---|---|---|---|
| | Derm | PanDerm | Derm | PanDerm | Derm | PanDerm |
| Base NMC (from Table 1) | 64.75 | 57.95 | 67.56 | 49.27 | **68.74** | 44.51 |
| Base NMC + Norm. | 66.60 | 61.03 | **69.46** | **54.89** | 66.43 | 47.94 |
| Random Projection | 67.29 | 57.43 | 66.99 | 49.57 | 68.74 | 40.72 |
| Random Projection + Norm. | 66.11 | 62.06 | 66.20 | 54.38 | 68.46 | 47.27 |
| Hyperbolic Projection | **81.41** | **80.24** | 63.79 | 43.21 | 65.28 | 59.59 |
| Hyperbolic Projection + Norm. | 80.15 | 80.15 | 60.08 | 53.91 | 64.77 | **63.73** |
| PCA | 64.75 | 56.67 | 67.26 | 50.23 | 68.74 | 44.51 |
| PCA + Norm. | 66.60 | 59.96 | 69.46 | 54.89 | 66.10 | 47.94 |
| LDA | 51.88 | 78.91 | 69.38 | 37.12 | 45.23 | 38.56 |

**Table 3.** Performance evaluation (balanced accuracy %) of our FM-based baselines using a general-purpose foundation model (CLIP ViT-L/14). Green and blue cells denote the best and second-best results, respectively.

| Method | HAM | DMF | D7P |
|---|---|---|---|
| **Our Baselines** | | | |
| FM with MLP | **88.38** | **90.43** | **71.19** |
| FM with NMC | 53.53 | 70.13 | 46.01 |
| **NMC Classifier Variants** | | | |
| Base NMC + Norm. | 55.11 | 71.26 | 46.52 |
| Random Projection | 52.15 | 69.99 | 43.79 |
| Random Projection + Norm. | 51.56 | 69.87 | 43.63 |
| Hyperbolic Projection | 80.05 | 53.50 | 59.59 |
| Hyperbolic Projection + Norm. | 80.05 | 57.20 | 60.10 |
| PCA | 53.53 | 70.13 | 45.86 |
| PCA + Norm. | 55.10 | 71.25 | 46.37 |
| LDA | 73.04 | 61.82 | 28.57 |

PanDerm accuracy on D7P from 44.51% to 63.73%, yielding the strongest NMC results overall. While both the hyperbolic projection and the MLP classifier involve learnable parameters, they differ fundamentally. The MLP learns flexible, general mappings from embeddings to class predictions, requiring more extensive training. In contrast, the hyperbolic projection embeds data in a geometric space that models hierarchies, enhancing clustering and interpretability with fewer parameters and less risk of overfitting. We finally assess classical dimensionality reduction techniques. Principal component analysis (PCA), which preserves variance without explicitly optimizing class separability, does not improve performance, whereas Linear discriminant analysis (LDA), designed to maximize between-class variance, delivers mixed, unstable results: while it achieves 78.91% on HAM with PanDerm, its performance deteriorates on other datasets due to the high intra-class variance. In summary, we conclude that normalization (as a non-learnable enhancement) and the hyperbolic projection (as a learnable enhancement) provide the most effective improvements to the NMC, each helping to narrow the gap to the MLP classifiers reported in Table 1 on different datasets.

**2. General-Purpose vs. Domain-Specific FMs.** To assess the importance of domain specialization, we repeat our experiments using a general-purpose FM—CLIP ViT-L/14 pretrained on natural images, replacing the dermatology-

specific model. Results are shown in Table 3. Despite lacking domain-specific pretraining, CLIP embeddings remain highly effective for parametric classifiers: the MLP achieves 88.38%, 90.43%, and 71.19% on HAM, DMF, and D7P, respectively, outperforming all prior `CL` methods. This supports our central claim: strong, transferable `FM` features, regardless of domain, can improve performance in `CIL`. In contrast, NMC variants suffer significant degradation. The base NMC achieves only 53.53% on HAM and 46.01% on D7P, far below its dermatology-initialized counterpart. Interestingly, while normalization and hyperbolic projection again improve performance (e.g., HAM jumps from 53.53% to 80.05%), they cannot fully bridge the gap, and their gains are inconsistent across datasets. Hyperbolic projection + normalization achieves a strong 60.10% on D7P but still trails the MLP by more than 11%. LDA continues to show erratic behavior: while it produces 73.04% on HAM (competitive with more structured NMC variants), it collapses entirely on D7P (28.57%), underscoring its sensitivity to class imbalance and feature distributions. Overall, these findings reinforce two observations: (1) parametric models like MLPs can extract meaningful decision boundaries from general-purpose `FMs`, making them highly effective for `CIL`; and (2) for other approaches like NMC that lack task-specific adaptation, alignment between the pretraining and target domain remains crucial.

## 5   Conclusions

This work demonstrates the clear advantage of leveraging frozen foundation models as class-incremental learners in dermatological image classification. Through systematic evaluation across three skin lesion benchmarks, we show that a simple approach, which is training a lightweight MLP on top of a frozen dermatology-specific backbone, can surpass upper-bound reference methods, without requiring complex regularization, replay, or architectural modifications. Remarkably, this MLP-based strategy maintains strong performance when built on general-purpose models like CLIP ViT-L/14, further reinforcing the value of rich, pretrained features in `CL` for medical applications. These findings yield three key insights. First, foundation models should be considered the default starting point for future research in `CL`. Second, nearest-mean classifiers still benefit substantially from domain-specific pretraining due to their limited representational flexibility. Third, our results emphasize the importance of aligning model design with the geometric properties of the embedding space. Specifically, incorporating inductive biases, such as learnable hyperbolic projections, can significantly close the gap between simple prototype-based classifiers and other learnable models while offering greater simplicity and interpretability. Taken together, we hope this work encourages the community to rethink the foundations of `CL`; i.e., shifting from building methods from scratch toward designing smarter, lighter learning systems that build on the strengths of powerful pretrained models. A promising future direction is to explore dynamic backbone adaptation and task-aware prompt tuning to further improve flexibility while retaining the benefits of strong pretrained representations.

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
