# OpenReview forum: "Foundation Models as Class-Incremental Learners for Dermatological Image Classification"
_MICCAI.org/2025/Workshop/MSB_EMERGE — MSB EMERGE 2025 Oral_

### Official Review · Reviewer_3T5a · 2025-07-04

**Clarity:** The paper is exceptionally well-writt…
**Recommendation:** 5
**Confidence:** 4

**Feedback:**

- Adding more classification metrics and general models would be beneficial in future work

**Justification:**

The authors addressed in this paper an interesting question for the community and evaluated it using an extensive and clear experimental setup.

**Reproducibility:**

Sufficient amount of details available for reproducing the main results, but open access is not provided to source code and/or data

**Strengths:**

-	**Extensive benchmark:** The methods were evaluated on three publicly available datasets against a lot of baselines.
-	**Relevant ablation studies:** The authors showed interesting findings for the community thanks to two relevant ablation studies.
-	The experimental setup is really clear, making the paper reproducible

**Summary:**

In this paper, the authors evaluate whether frozen dermatology foundation models can classify skin lesions efficiently in a class incremental setting. They demonstrate their benefit against classical continual learning methods in an extensive benchmark (3 public datasets) without any forgetting.

**Weaknesses:**

No major weaknesses

---

### Official Review · Reviewer_Kkc3 · 2025-07-07

**Recommendation:** 5
**Confidence:** 4

**Clarity:**

The paper is clear and well-written, with minor areas for improvement in clarity

**Feedback:**

- There is a line repeated in page 5: "However, their performance lag behind their MLP counterparts"

**Justification:**

The paper presents an interesting approach to leveraging foundation models for dermatological image classification with minimal training. It is well written and addresses a problem of high relevance to the medical imaging community.

**Reproducibility:**

Sufficient amount of details available for reproducing the main results, and open access is provided (or promised upon acceptance) to source code and/or data

**Strengths:**

- **Strong performance**: The proposed methods achieve competitive results compared to state-of-the-art continual learning approaches. In particular, the variant using a lightweight MLP classifier outperforms all baseline methods with only minimal training.
- **Comprehensive evaluation**: The methods are evaluated across three publicly available datasets, supporting the generalizability of the results.
- **Thorough experimentation**: The ablation studies on the nearest-mean classifier variant are comprehensive, demonstrating several strategies to enhance performance without requiring additional training.

**Summary:**

The paper investigates the use of foundation models (FMs) as a starting point for continual learning (CL) in dermatological image classification. The results show that features extracted from frozen FMs can outperform standard CL methods, with performance generalizing well across three publicly available datasets.

**Weaknesses:**

- **Lack of technical novelty**: The approach builds on existing models with only minor adaptations. However, given the paper’s focus on leveraging foundation models to reduce the need for costly training and evaluation, this is not a significant limitation.

---

### Official Review · Reviewer_XG7u · 2025-07-09

**Recommendation:** 5
**Confidence:** 4

**Clarity:**

The paper is clear and well-written, with minor areas for improvement in clarity

**Feedback:**

A brief discussion of limitations or potential directions for future work would help round out the contribution.

**Justification:**

The method is simple yet effective, and the results suggest that foundation model-based approaches may be a promising direction for class-incremental learning in medical domains.

**Reproducibility:**

Sufficient amount of details available for reproducing the main results, and open access is provided (or promised upon acceptance) to source code and/or data

**Strengths:**

* The method is evaluated across multiple datasets and foundation models, demonstrating strong generalizability.
* It achieves strong performance compared to existing continual learning approaches.
* The ablation study on NMC variants is thorough and provides valuable insights.

**Summary:**

This paper explores class-incremental learning in dermatology by leveraging foundation models with lightweight MLPs or nearest mean classifiers, showing that these simple strategies can be highly effective.

**Weaknesses:**

* The overall approach does not seem particularly novel; however, the strong empirical results compensate for this.